# ChatGPT-Powered Hierarchical Comparisons for Image Classification

Zhiyuan Ren, Yiyang Su, and Xiaoming Liu

Department of Computer Science and Engineering, Michigan State University
East Lansing, MI 48824
{renzhiy1, suyiyan1, liuxm}@msu.edu

## Abstract

The zero-shot open-vocabulary setting poses challenges for conventional image classification. Vision-language models pretrained on image-text pairs like CLIP offer a solution based on comparing image and class label embeddings. Incorporating class-specific knowledge provided by large language models (LLMs) such as ChatGPT in descriptions can further enhance CLIP's accuracy. However, CLIP still exhibits a bias towards certain classes and generates similar descriptions for closely related but different classes. To address these problems, we present a novel image classification framework via hierarchical comparisons. By recursively comparing and grouping classes with LLMs, we construct a class hierarchy. With such a hierarchy, we can classify an image by descending from the top to the bottom of the hierarchy, comparing image and text embeddings at each level. Through extensive experiments and analyses, we demonstrate that our proposed approach is intuitive, effective, and explainable. Code is available here.

## 1 Introduction

Conventional image classification approaches typically evaluate their performance on the same set of categories as their training data. However, this evaluation paradigm fails to capture the challenges in real-world scenarios, where classes in the test set are not overlapped with the training set, *i.e.* zero-shot, and the test set may comprise an unknown number of classes, *i.e.* open-vocabulary. Zero-shot open-vocabulary image classification, on the other hand, accommodates the open-ended nature of class diversity in practical applications, thereby serving as a more accurate proxy for real-world generalizability, and is the focus of this paper.

Through training with aligned image-text pairs, CLIP [37] can generate embeddings for both seen and unseen classes. To apply it in zero-shot open-vocabulary image classification, one simply obtains an image embedding from CLIP's image encoder and compares it with the text embeddings of all label candidates. Recently, Menon and Vondrick [29] propose a simple yet effective method, classification by description. They use LLMs to generate text descriptions for unseen class labels and compare the image embedding with the embeddings of the descriptions. Their method exploits the semantic information offered by LLMs to bolster the efficacy of CLIP in image classification.

Nevertheless, their classification accuracy is still limited by a few factors. Firstly, in image classification, CLIP is biased towards some classes in classification [45]. In practice, we observe that CLIP fails to distinguish among semantically similar categories, and often misclassifies a few classes as one specific class. For instance, in the CUB [43] dataset, the great-crested flycatcher, the least flycatcher, and the yellow-bellied flycatcher are misclassified by CLIP as Arcadian flycatcher most of the time. Classification by description [29] does not address this issue because the descriptions for these classes are similar. Secondly, descriptions generated by LLMs just from the labels are generic

37th Conference on Neural Information Processing Systems (NeurIPS 2023).

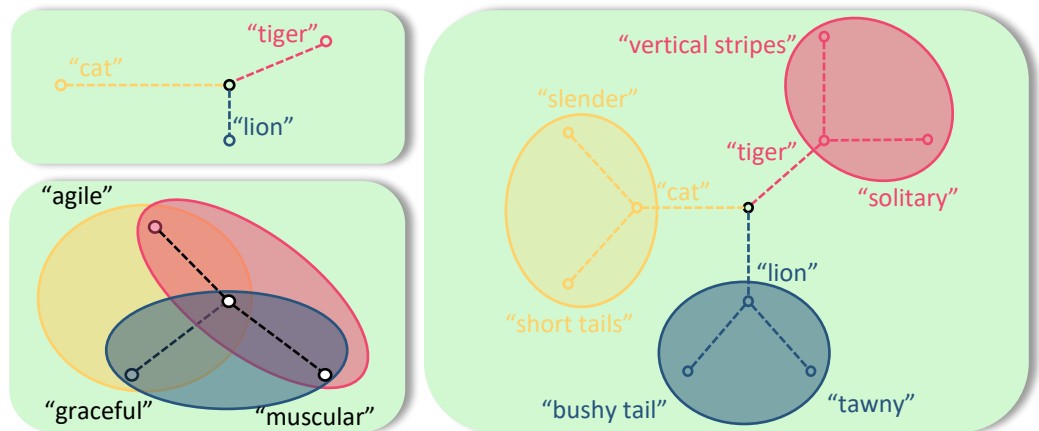

**Figure 1:** A toy example of classifying a felid. Top left: CLIP [37] only uses the class labels. Bottom left: Menon and Vondrick [29] expand the space of each class by using text descriptions. But their text descriptions are not dependent on the other class so their descriptions are suboptimal for classification. Right: We generate text prompts by comparing different classes to reduce inter-class similarity and make the decision boundary of each class more compact.

and unsuitable for discriminating a class from its semantic neighbors. As shown in Fig. 1, the LLM generates "agile" as the description for the classes `cat` and `tiger`, "graceful" for `cat` and `lion`, and 'muscular' for `lion` and `tiger`. These descriptions are not helpful for classifying images of a cat-like animal because all of them are shared by two classes.

In light of these problems, we propose a novel framework for zero-shot open-vocabulary image classification. Inspired by how human beings hierarchically organize visual and linguistic concepts in the open world, our method organizes the classes in the test set into a hierarchy with the help of LLMs. A hierarchy is preferable for classification because different levels of classification require different features or descriptions. Telling a tiger from a car is nowhere near telling a tiger from a cat. Therefore, when we build the hierarchy and perform inference, we only compare classes or groups of classes within the level or nodes where they belong.

Building the hierarchy without human interaction is possible because the state-of-the-art LLMs have common sense and some expert knowledge in, *e.g.*, ornithology. In our approach, under the scope of wrens, LLMs describe that a house wren has "a reddish-brown coloration with fine streaking on their back and wings" and a winter wren has "a dark brown coloration with a mottled appearance on their back and wings". With such discriminative text descriptions for each class, when differentiating between house wrens and winter wrens, the CLIP classifier will specifically examine the color tone of the bird and the texture on the birds' backs and wings, which is both intuitive and logical. Additionally, we will be able to explain the grounds on which the classifier makes decisions by looking at the descriptions with the highest score at each level of the hierarchy.

Through comprehensive evaluations, we show that our performance is better than the existing method, Menon and Vondrick [29], and the vanilla CLIP [37] image classifier across various backbones and datasets, which showcases the effectiveness of the proposed hierarchical comparison approach.

In summary, the contributions of this paper include

- ✔ We leverage prior knowledge of the LLMs to construct knowledge trees by iterative grouping and comparing. Combined with CLIP [37], we can obtain a hierarchical and comparative text representation.
- ✔ We achieve state-of-the-art performance in zero-shot image classification on different datasets and backbone settings. Without additional training, we can gain appreciable increases of up to 10%.
- ✔ Moreover, we show the strong explainability of our model brought by the hierarchical descriptions. It can tell why the model chooses a category as the prediction and at which layer of description, the model makes the choice.

## 2 Related Work

**LLMs.** Recently, LLMs [1, 3, 12, 19, 20, 25, 34, 40, 46, 49] have witnessed significant advances, revolutionizing natural language processing tasks with hundreds of billions of parameters and special tricks such as Reinforcement Learning from Human Feedback. Researchers have started to harness the generation powers of LLMs. Frozen [41] and PICa [47] apply LLMs in visual question answering. Similar to our approach, Menon and Vondrick [29] directly generate textual descriptions from labels to assist language-vision models in image classification. However, as discussed above, their approach suffers from a lack of discriminability among semantically similar classes.

Contrarily, we embed ChatGPT into a hierarchical framework for image classification, starting at a high or coarse level (*e.g*., animals *vs*. vehicles) and going down into the realm of low-level or fine-grained classification (*e.g*., a house wren *vs*. a winter wren). ChatGPT will be prompted to compare object classes at different levels so that it can focus on different class-defining aspects at the respective level.

**Language-Vision Pretraining.** CLIP [6, 14, 37] and other vision-language models (VLMs) manage to learn a single embedding space for both image and text using contrastive learning, bridging the gap between the two modalities. CLIP has been widely adopted to use text to guide image [8, 32, 35] and 3D generation [17, 30, 38]. In particular, with an image and a textual description of the target class as input, CLIP can associate visual and textual representations to make predictions without explicitly training on that specific class. This enables classifying images into unseen classes based on semantic relationships between images and text. In our approach, we take advantage of CLIP to map both the input image and GPT-generated distinguishing texts to their embedding space.

**Text Prompting.** Text prompting can be generally divided into soft prompting and hard prompting. Compared to soft prompting [4, 5, 18, 21, 22, 36], which refers to using a learnable token as the prompt, hard prompting [53] gets prompts by filling phrases into preset templates. Soft prompts have demonstrated the ability to achieve commendable performance in image classification [13, 51, 52]. However, hard prompts could be preferable because they do not require any training and offer better explainability. Our approach belongs to the group of hard prompts, thus inheriting the advantages associated with such prompts while not sacrificing performance compared to soft prompts.

**Hierarchical Representations.** Hierarchical structures [9, 15, 23, 24, 44] refer to the organization of visual information in a hierarchical manner, where higher-level concepts or features are built upon lower-level ones. It captures the inherent hierarchical structure and relationships present in real-world tasks, allowing for more efficient and discriminative processing. This concept draws inspiration from the human brain's hierarchical organization [28], where lower-level neurons form simple patterns, which are then combined to form higher-level representations. HGR-Net [48] organizes the classes with the help of WordNet [31] while we exploit the prior knowledge of LLMs to produce descriptions based on other classes. Plus, their network needs to be trained but we only require pretrained VLMs. Like our approach, CHiLS [33] also uses LLMs and does not require training, but relies solely on hierarchical labels and does not leverage class-specific descriptions or comparisons.

## 3 Methodology

### 3.1 Problem Definition

Given an input image $\mathbf{I}$, we attempt to assign it to the correct category $c_i$ among an arbitrary set of categories $\{c_1, c_2, c_3, ..., c_n\}$, on which the model has not been trained for classification. One of the most common methods [37] is to use a pretrained VLM to map the image and each class to the aligned latent space and select the pair that has the highest cosine similarity with the image embedding,

$$\mathbf{x} = E_v(\mathbf{I}), \tag{1}$$

$$\mathbf{t}_i = E_t(c_i), \tag{2}$$

$$\text{pred} = \arg\max_i (\mathbf{x} \cdot \mathbf{t}_i^T). \tag{3}$$

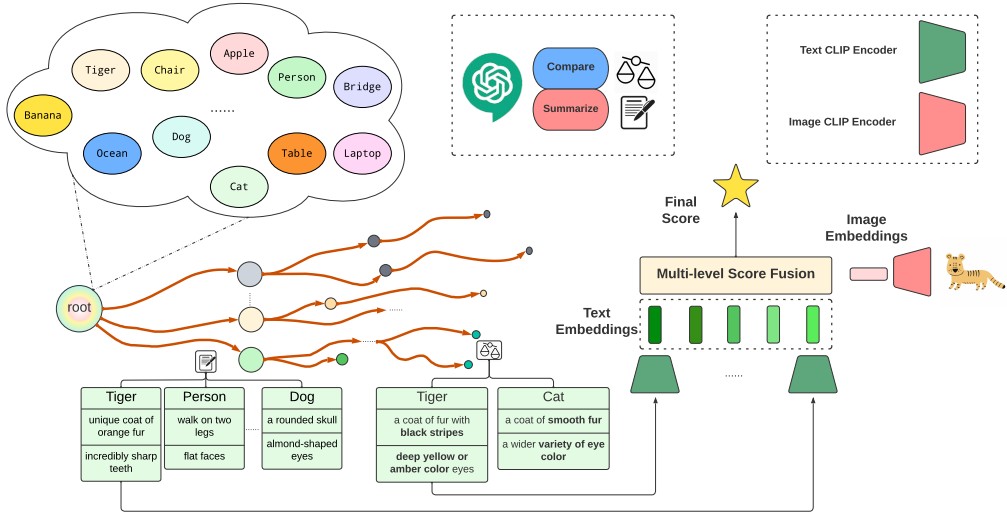

**Figure 2: Illustration of our architecture.** Given a set of labels, we alternately employ grouping and LLM's generated descriptions to build knowledge trees. Grouping provides a more detailed scope for generating descriptions, and generating descriptions provides more distinguished embeddings for grouping. Upon building the knowledge tree, we compute the similarity score between the hierarchical description and the image embedding to obtain the final classification score.

where $E_v$ and $E_t$ are the vision encoder outputting the image feature $\mathbf{x}$ and the text encoder outputting the text feature $\mathbf{t}_i$, respectively. In Eq. 2, the text encoder can take either a simple template-based prompt like "a photo of {category name}" [37] or a learnable prompt [13].

In spite of the promising results of the existing prompt generation methods, we aim to leverage the abundant prior knowledge in LLMs to build a class hierarchy. The overall approach is inspired by how people perceive and recognize objects in the real world where classes are hierarchically grouped and compared to each other.

## 3.2   Building Knowledge Tree by Grouping and Comparing

To recognize objects in the real world, humans often follow a coarse-to-fine and hierarchical approach. For example, to recognize a husky dog, one can follow a classification process by grouping it into the animal kingdom. Subsequently, based on specific characteristics such as the shape of its nose and the texture of its fur, it can be further classified into the dog family. Finally, within the dog family, a feature comparison can be conducted with other dogs to identify their specific breed or category.

Within this cognitive process, two fundamental operations play a pivotal role: grouping and comparing. In the proposed methodology, we initially consolidate similar categories into one unified group. Subsequently, we leverage the knowledge derived from the LLM to acquire comparative descriptions pertaining to distinct categories within the same group. Building upon the improved features, we can then perform a successive round of grouping, iteratively executing the above steps. The architecture is shown in Fig. 2.

Specifically, following [29], we utilize ChatGPT to generate initial descriptions for each object under consideration and employ the pretrained CLIP text encoder to map these descriptions into the latent space. Using these features, we invoke a clustering algorithm, *e.g.*, $k$-means [27], for grouping. Even with extensive prior knowledge of LLMs, comparing a large number of categories ($> 10$) presents a formidable challenge, as it needs to generate a comprehensive description matrix that grows quadratically with the number of categories. To address this issue, we employ tailored strategies for generating descriptions based on the group size, allowing for more efficient and effective comparisons. We adopt summary-based comparison for larger groups and direct comparison for smaller ones.

**Summary-based Comparison.** Given the substantial number of categories involved, we can leverage the capabilities of the LLM to summarize the overarching characteristics of the group (see example below). By incorporating these summarized characteristics into a new query, we can generate a comparative description that captures the distinctive features and relationships among the categories. This approach enhances our ability to effectively compare and contrast the elements within a group.

```
Q: Summarize the following categories with one sentence: {category list}?

A: {subset description}

Q: What are useful features for distinguishing {category name} in a dataset: {subset description}?

A: -
```

**Direct Comparison.** When the number of elements within a group is relatively small, a direct query to the LLM enables us to obtain an exceptionally comparative description (see example below). This approach capitalizes on the LLM's capabilities to provide a comprehensive and insightful analysis, enhancing the comparative understanding of the elements within the group.

```
Q: What are useful features for distinguishing a {target category} from {other categories} in a photo?

A: -
```

Utilizing these detailed and comparative descriptions, we can achieve more nuanced text embedding. Subsequently, we can continue to perform clustering, followed by additional comparisons to derive new descriptors.

As shown in Fig. 3, this process is recursively executed until the number of leaf nodes becomes fewer than a predefined threshold. At this point, we successfully generate the entire knowledge tree. Within a narrower group range, we utilize queries to the LLM to compare classes within the group, generating more comparative descriptions. Unlike [29], our method ensures that the description is adaptive to different class sets.

```
# T_init[n, d]              - minibatch of initial text embeddings
# Text_encoder              - CLIP Text Encoder
# LLM(prompt)->description   - ChatGPT model

# Target: collect tree-level description

def build_tree_in_loop(T_emb):
    new_groups = K-means(T_emb)

    for group in new_groups:
        if len(group) == 1:
            # Skip processing single-item group
            pass

        elif len(group) > thres:
            # Generate more fine-grained descriptions
            summary = LLM("summarize the following categories(...)")
            descriptions = LLM(group, summary)
            collect(descriptions)

            # Keep Grouping into smaller ones
            T_new = Text_encoder(description)
            build_tree_in_loop(T_new)

        elif 1 < len(group) <= thres:
            # Generate comparative descriptions
            descriptions = LLM("compare the following categories(...)")
            collect(descriptions)

# Starting point
build_tree_in_loop(T_emb=T_init)
```

**Figure 3:** Psuedo-code for building the knowledge trees.

### 3.3 Knowledge Tree Traversal and Multi-level Score Fusion

After building the knowledge tree, we can have text features at multiple levels of granularity, which are represented as $D = \{\mathbf{D}_1, \mathbf{D}_2, \ldots, \mathbf{D}_i, \ldots, \mathbf{D}_N\}$, where $N$ is the number of test classes, $\mathbf{D}_i \in \mathbb{R}^{M_i \times C}$ is the hierarchical text embeddings for category $i$, $C$ is the dimension of the text embedding, and $M_i$ is the number of comparative descriptions for category $i$ depending the tree structure. Given the image feature $\mathbf{x}$, the fused score $s$ can be expressed as:

$$q(j) = \mathbf{x} \cdot (\mathbf{D}_i)^j \,, \tag{4}$$

$$r(\mathbf{x}, i) = \frac{q(1) + \sum_{j=2}^{M_i} q(j) \prod_{k=1}^{j-1} \mathbb{1}[q(k+1) > q(k) + \tau]}{1 + \sum_{j=2}^{M_i} \prod_{k=1}^{j-1} \mathbb{1}[q(k+1) > q(k) + \tau]}, \tag{5}$$

$$s(\mathbf{x}, i) = (1 - \lambda) (\mathbf{x} \cdot \mathbf{t}_i) + \lambda (r(\mathbf{x}, i)). \tag{6}$$

Here, $q(j)$ calculates the cosine similarity between the image embedding $\mathbf{x}$ and the $j$-th text embedding of category $i$. $r(\mathbf{x}, i)$ calculates the running average of the longest sequence of monotonically

increasing $q$ values. The final similarity score $s(\mathbf{x}, i)$ is a weighted average of the cosine similarity between the image embedding $\mathbf{x}$ and the $i$-th class label embedding $\mathbf{t}_i$ and $r(\mathbf{x}, i)$.

In this fusion method, the new score is merged only when encountering a higher score in the subsequent node compared to the current node. This addresses the ambiguity issue that arises from excessively detailed descriptions. For instance, when comparing a yellow sparrow and a red sparrow, the most comparative description might include a specific detail such as "yellow coloration across the full body". However, such a detailed description could inadvertently increase the scores of unrelated objects like a banana or wood. To mitigate this issue, our fusion method ensures that the subsequent scores will be dropped if the current description has a low score, such as "a short and wide bill" for a banana or wood. Besides, our fusion method takes into account the score of the initial description as an offset, and we introduce a constant hyperparameter $\lambda \in [0, 1]$ to balance the two terms according to Eq. 6; $\tau$ is the tolerance for score reduction, which is close to 0.

## 4 Experiments

### 4.1 Implementation Details

There are four hyperparameters in our method, including the number of groups $N$ in the $k$-means algorithm [27], the threshold $l$ for leaf nodes, the weight $\lambda$ assigned to score offset, and the tolerance $\tau$ for score reduction. $N$ exhibits an inverse proportionality relationship with the number of labels and $\lambda$ is roughly determined by the inter-class similarity of the dataset. We generally set $l$ to 2 or 3 and $\tau$ to 0. In our approach and baseline setting, we employ CLIP ViT-L/14 for encoding the initial text embedding and leverage ChatGPT to handle all text completion tasks. On a single Nvidia RTX A6000 GPU, it is feasible to replicate all the results of our paper within approximately two hours.

### 4.2 Quantitative Results

Unlike many prior works that require fine-tuning for downstream tasks, our approach eliminates the need for any training in the zero-shot setting. Instead, we simply enrich the text information by incorporating a hierarchical description for each category. In line with the methodology employed by [29], we expand the ImageNet validation by introducing two new categories, each containing five additional images.

We conduct experiments on six different image classification benchmarks, *i.e.* ImageNet [10], CUB [43], Food101 [2], Place365 [26], Oxford Pets [50], and Describable Textures [7], using two baseline models: CLIP [37] and OpenCLIP [6]. For CLIP, we evaluate its performance with four backbones (Res-50 [16], ViT-B/32, ViT-B/16, and ViT-L/14 [11]), while for OpenCLIP, we use three backbones (ViT-B/16, ViT-L/14, and ViT-G/14). When generating the class hierarchies, we fix the ratios of the number of clusters to the class number of each dataset throughout all experiments.

As shown in Tab. 1, our proposed hierarchical comparison framework can improve upon the baseline methods of image classification across various datasets and backbones. This demonstrates that our prompts generated from comparing classes hierarchically are more suitable for zero-shot image classification. Moreover, it is evident that our approach is robust and generalizable to different zero-shot image classification settings.

There is an isolated exception - the proposed approach does not improve over OpenCLIP [6] on the Food101 dataset. Although, we note that our approach outperforms OpenCLIP as its baseline on all other datasets and CLIP on Food101. We speculate that since OpenCLIP uses different training data from CLIP, its image and text encoders may generate indistinguishable embeddings for foods. In fact, following the reported results in [39], Food101 is the one of few datasets where CLIP has a higher accuracy than OpenCLIP under the same backbone, which justifies our speculation to some extent.

On the CUB dataset, the margin between our approach and the baseline is relatively smaller. We attribute this to the fact that both CLIP and OpenCLIP lack relative perception capability. Their encoders, while being sensitive to features such as colors and textures, cannot comprehend the comparatives. The semantic difference between large and larger, long and longer, and yellow and darker yellow is not reflected in their embeddings. This phenomenon in turn stems from the text-image pairs used for training CLIP and OpenCLIP. As the CUB dataset contains exclusively birds, the differences between classes are more fine-grained. Consequently, the comparisons would more

**Table 1:** Zero-shot classification accuracy gains over CLIP and OpenCLIP category name embedding baseline. We see significant increases across all the settings except Food101 with the OpenCLIP.

| Dataset | ImageNet | | | CUB | | | Food101 | | |
|---|---|---|---|---|---|---|---|---|---|
| Architecture | CLIP | Ours | Δ | CLIP | Ours | Δ | CLIP | Ours | Δ |
| Res-50 | 54.85 | 60.63 | +5.78 | 49.02 | 49.86 | +0.74 | 71.61 | 75.44 | +3.83 |
| ViT-B/32 | 58.47 | 63.88 | +5.41 | 52.23 | 54.18 | +1.95 | 78.83 | 83.02 | +4.19 |
| ViT-B/16 | 63.53 | 68.95 | +5.42 | 56.66 | 59.25 | +2.59 | 86.17 | 88.13 | +1.96 |
| ViT-L/14 | 76.60 | 79.64 | +3.04 | 63.46 | 65.45 | +1.99 | 91.86 | 93.17 | +1.31 |

| Dataset | Places365 | | | Oxford Pets | | | Describable Textures | | |
|---|---|---|---|---|---|---|---|---|---|
| Architecture | CLIP | Ours | Δ | CLIP | Ours | Δ | CLIP | Ours | Δ |
| Res-50 | 32.48 | 36.83 | +4.35 | 74.60 | 79.99 | +5.39 | 37.63 | 45.67 | +8.04 |
| ViT-B/32 | 37.23 | 40.73 | +3.50 | 79.80 | 82.69 | +2.89 | 40.82 | 49.19 | +7.37 |
| ViT-B/16 | 38.21 | 41.52 | +3.31 | 81.68 | 87.19 | +5.51 | 43.56 | 51.26 | +7.70 |
| ViT-L/14 | 38.65 | 41.13 | +2.48 | 87.92 | 92.84 | +4.92 | 51.01 | 58.04 | +7.03 |

| Dataset | ImageNet | | | CUB | | | Food101 | | |
|---|---|---|---|---|---|---|---|---|---|
| Architecture | OpenCLIP | Ours | Δ | OpenCLIP | Ours | Δ | OpenCLIP | Ours | Δ |
| ViT-B/16 | 62.96 | 67.08 | +4.12 | 66.81 | 68.20 | +1.39 | 84.85 | 84.59 | -0.26 |
| ViT-L/14 | 68.34 | 72.69 | +4.35 | 75.32 | 75.80 | +0.48 | 89.40 | 89.35 | -0.05 |
| ViT-G/14 | 70.94 | 75.90 | +4.96 | 84.19 | 85.19 | +1.00 | 93.41 | 92.68 | -0.73 |

| Dataset | Places365 | | | Oxford Pets | | | Describable Textures | | |
|---|---|---|---|---|---|---|---|---|---|
| Architecture | OpenCLIP | Ours | Δ | OpenCLIP | Ours | Δ | OpenCLIP | Ours | Δ |
| ViT-B/16 | 38.80 | 42.52 | +3.72 | 84.03 | 87.47 | +3.44 | 46.72 | 54.01 | +7.29 |
| ViT-L/14 | 38.46 | 44.12 | +5.66 | 87.13 | 89.62 | +2.49 | 51.69 | 61.79 | +10.10 |
| ViT-G/14 | 41.20 | 45.61 | +4.41 | 92.37 | 95.07 | +2.70 | 63.70 | 69.69 | +5.99 |

**Table 2:** Comparsion with the zero-shot SoTA method under fair settings. Overall, we gain ∼1% improvement across different backbones and datasets.

| Backbone | Method | ImageNet | CUB | Food101 | Place365 | Pets | Texture | Mean |
|---|---|---|---|---|---|---|---|---|
| ResNet-50 | [29] | 60.20 | 49.10 | 74.94 | 35.79 | 78.58 | 44.63 | 57.20 |
| | Ours | **60.63** | **49.86** | **75.44** | **36.83** | **79.99** | **45.67** | **58.07** |
| ViT-B/32 | [29] | 63.48 | 53.80 | 82.79 | 40.15 | 81.60 | 45.69 | 61.25 |
| | Ours | **63.88** | **54.18** | **83.02** | **40.73** | **82.69** | **48.19** | **62.12** |
| ViT-B/16 | [29] | 68.43 | 58.85 | 87.68 | 40.87 | 86.56 | 49.15 | 65.25 |
| | Ours | **68.95** | **59.25** | **88.13** | **41.52** | **87.19** | **51.26** | **66.05** |
| ViT-L/14 | [29] | 75.36 | 65.02 | 93.00 | 40.13 | 92.10 | 55.95 | 70.26 |
| | Ours | **79.64** | **65.45** | **93.17** | **41.13** | **92.84** | **58.04** | **71.71** |

often rely on comparing the same attributes, which is not one of the strengths of CLIP or OpenCLIP. For more experiments and analyses please refer to the supplementary materials.

Additionally, we compare our method to the state-of-the-art zero-shot approach. To ensure a fair comparison, we employ the same model weights for each distinct backbone setting. Moreover, we utilize the same LLM, ChatGPT [3], to generate descriptions, where the different factor lies in our hierarchical description structure. As shown in Tab. 2, our approach can consistently outperform the current state-of-the-art method in zero-shot classification.

Notably, we surpass the state of the art [29] on Describable Textures by a considerable margin. This is because, without the information of other classes as the context, LLMs generate class descriptions that lack discriminability. ChatGPT [3] characterized the `gauzy` class as "a thin, translucent fabric" with "a light, airy feel" and "a delicate or lacy appearance" and the `lacelike` class as an "intricate, detailed pattern" "often used for decoration". Even though correct, these descriptions do not provide any discriminative information, which renders them ineffective in classifying the two textures. On the other hand, in our hierarchy, `gauzy` and `lacelike` are two leaf nodes with a common parent. When comparing them, our approach describes the `gauzy` class as "a more transparent or sheer appearance" and "a more delicate texture with a softer hand-feel" and the `lacelike` as "a lightweight

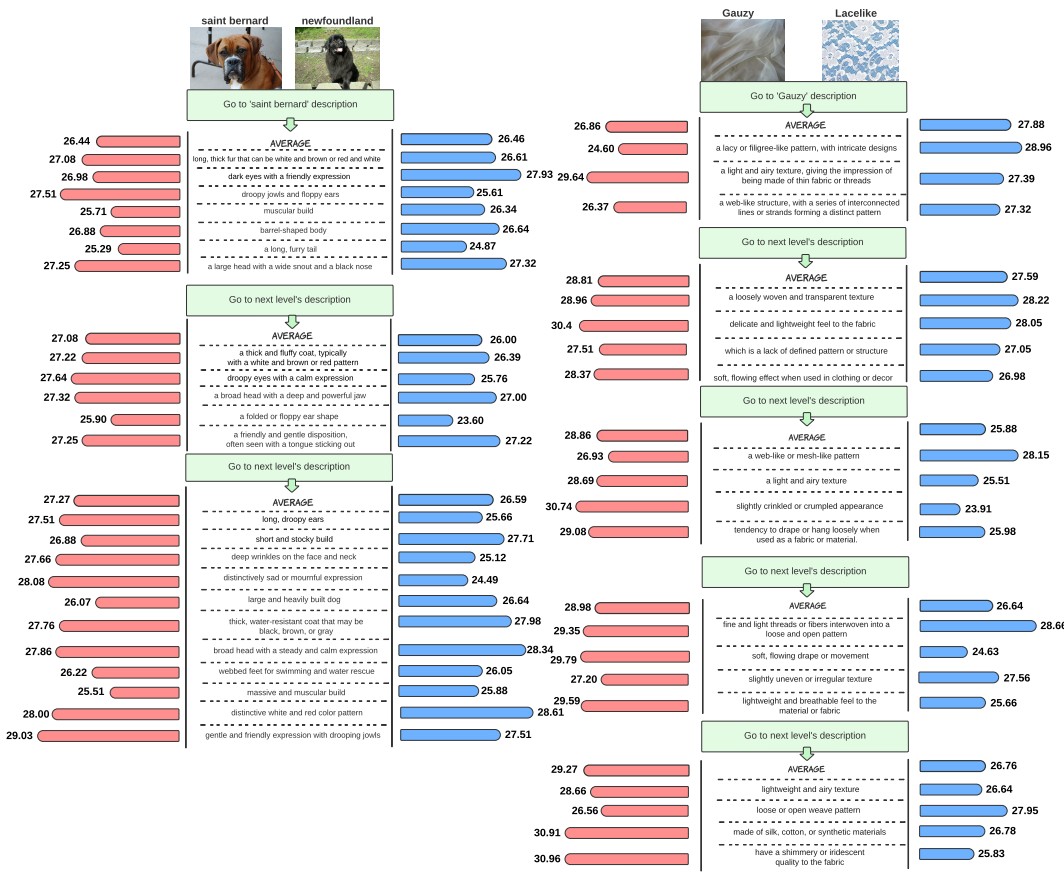

**Figure 4:** We show two inference examples and their score distribution by hierarchically querying the description with the image embedding. The description corresponds to the left input image and the right image is the sample from the confusing categories relative to the left one (zoom in for small fonts).

and almost transparent appearance" which has "visible holes or gaps throughout the material". Such descriptions accurately capture the subtle differences between these two materials and facilitate zero-shot classification between these two categories, which manifests in the superiority of our proposed hierarchical comparison approach for zero-shot image classification.

## 4.3 Interpretability and Analysis

Fig. 4 illustrates two groups of examples that demonstrate the inference and score calculation in our method. Specially, we input two images and compute their similarity to the hierarchical descriptions associated with the left image. Due to the weak comparative information in the early descriptions, similar images tend to yield similar scores at the beginning. As we progress towards a more comparative description, the disparity between the scores of the two images gradually increases. Besides, this also highlights the interpretability of our method, as it provides the user insights regarding the attributes that elicit stronger responses to the input. Additionally, it indicates the specific layer of description at which the scores of different images diverge, widening the gap.

In order to further investigate the factors contributing to the improved performance of our model, we conduct a comparison of the confusion matrices between our model and CLIP, using the same backbone. As shown in Fig. 5, the number of cases where Ragdolls is misclassified as Birman is reduced from 54 to 37, as contrasting descriptions bring out distinguished text features.

Fig. 6 illustrates the step-by-step progression of our comparative hierarchical description, pushing the image embedding closer to its corresponding ground-truth text embedding. Simultaneously, this process gradually distances the confusing label from the image embedding. We visualize this process using t-SNE [42] reduction to two dimensions, while also regularizing them into unit vectors.

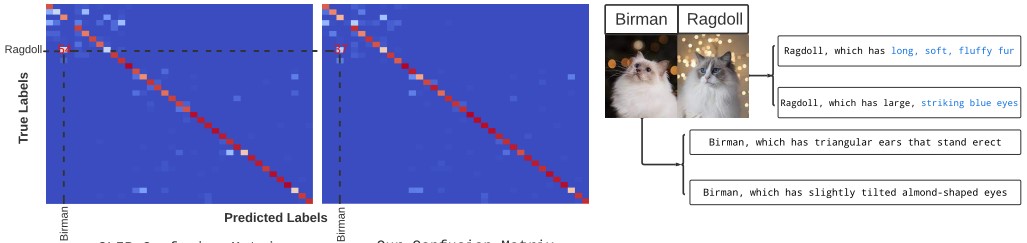

**Figure 5:** We highlight the difference between the confusion matrices at the position where Ragdoll samples are misclassified into Birman. At the right part, we show the comparative descriptions which explain where the gain comes from.

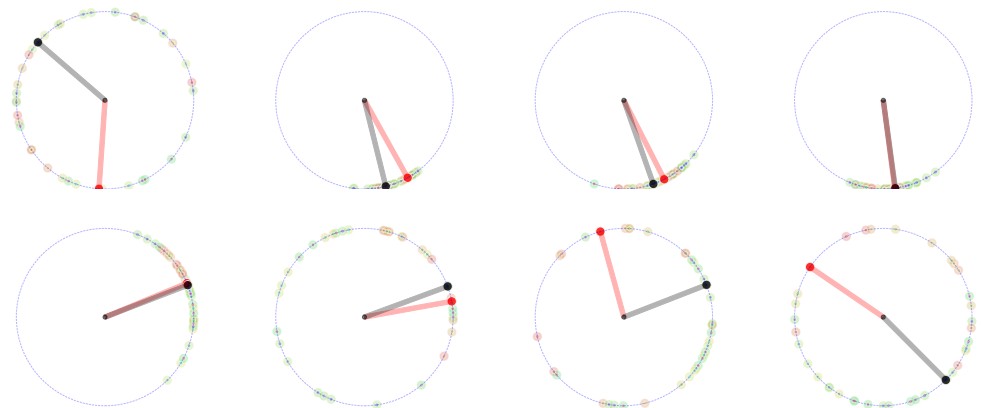

**Figure 6:** T-SNE visualization of similarity between the image embedding and the text embedding. Row 1 shows how our hierarchical descriptions help pull the image embedding closer to the groudtruth text embedding; Row 2 shows how our hierarchical descriptions push the confusing text embedding away from the current image embedding. ↗ represents the image embedding; ↗ represents the text embedding (groundtruth in Row 1; confusing category in Row 2); Other points represent the remaining text embeddings in the latent space.

## 4.4 Ablation Study

We ablate on four distinct hyperparameters: $\lambda$, $N$, depth, and $\tau$ utilizing a ViT-B/32 architectural foundation. Metrics derived from mean accuracy are employed for $\lambda$ and $\tau$ across datasets including ImageNet [10], CUB [43], Food101 [2], Place365 [26], Oxford Pets [50], and Describable Textures [7]. Alternatively, depth and $N$ are exclusively evaluated based on ImageNet [10] performance.

**Score Momentum $\lambda$.**    As illustrated in Fig. 7a, there exists a direct relationship between increasing $\lambda$ values and performance enhancement. However, solely relying on scores from the proposed methodology without offset utilization results in a decline in performance. Optimal performance is achieved when $\lambda$ is set to 0.9.

**Group Number $N$.**    Larger group numbers, such as 8, yield rudimentary hierarchies and generalized descriptions, whereas smaller values extend the hierarchy's development phase. Represented in Fig. 7b, it reveals that moderate increases in accuracy corresponded to fourfold evaluations. Inversely, a sixfold acceleration is observed, albeit at a cost of a $1.64\%$ dip in accuracy. Typically, $N$ oscillates between 4 to 6 and directly correlates with the class count.

**The Depth of the Prediction.**    Incorporating average aggregation is advantageous for score computation across variable depths. Results presented in Fig. 7c, when evaluated on ImageNet with ViT-B/32, underscore the superior performance of the next layer over its predecessor, attributed to the incorporation of divergent descriptions. A successive rise in performance is evident with depth augmentation.

**Tolerance for Score Reduction $\tau$.**    As delineated in Fig. 7d, the evolution of $\tau$ from 0 to 1 maintains nearly consistent performance, peaking at 1.0. Although optimal performance is observed at $\tau$=1, marginal variations across datasets necessitate the retention of this hyperparameter.

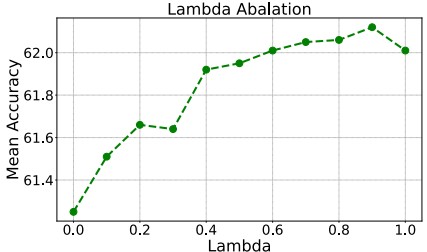

(a) **Ablation study of** $\lambda$. We fix $\tau$ as 0, use ViT-B/32 as the backbone, and compute the mean accuracy across six datasets.

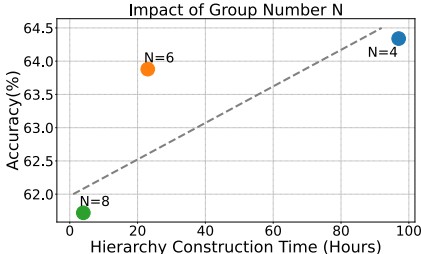

(b) **Ablation study of N.** An ablation on the ImageNet with ViT-B/32 for varying N values follows.

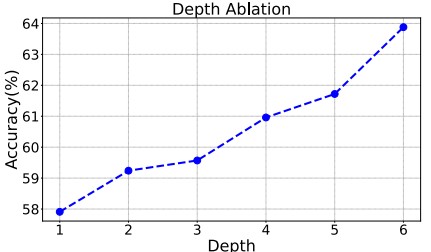

(c) **Ablation study of Depth.** Average aggregation is used in computing scores across various depths on ImageNet.

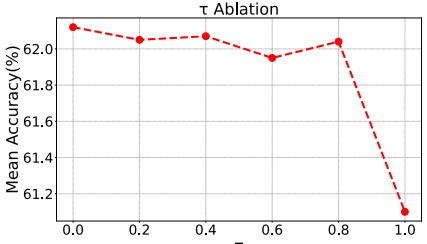

(d) **Ablation study of** $\tau$. We fix $\lambda$ as 0.9, use ViT-B/32 as the backbone, and compute the mean accuracy across six datasets.

**Figure 7:** Ablation study across different hyperparameters.

### 4.5 Failure Cases

By comparing confusion matrices, we find that there are indeed some classes whose classification accuracy decreases when using descriptions at greater depths. We pick representative classes and their descriptions of where they have lower classification accuracy than only using the previous depth.

We observe that when ChatGPT differentiates between closely related classes in its deeper layers, it often references attributes such as length and size, which can be challenging to represent visually. Consider the statement: *"Newfoundland dogs are typically larger than Chows. Male Newfoundlands can weigh up to 150 pounds and reach heights of 28 inches, whereas Chows generally weigh between 50-70 pounds and stand 20-25 inches tall."* The CLIP model's encoder, however, struggles to interpret this length information accurately due to variations in camera angles and positions. This observation suggests that detailed text descriptions might pose comprehension challenges for CLIP. Thus, enhancing the synergy between LLMs and CLIP presents itself as a promising avenue for future research.

## 5 Conclusion

In this work, we propose a training-free, explainable, and effective zero-shot image classification method with enriched hierarchical descriptions powered by LLMs, such as ChatGPT. The knowledge tree constructed by alternately grouping descriptions and generating descriptions allows us to achieve SoTA in different settings. A central theme of our work is to inspire the field by emphasizing that LLMs possess the potential to enrich semantics and make them useful for contemporary machine learning tasks. How to extract semantics worth expanding in the current machine learning tasks is the key to exploring the potential of LLMs.

**Limitations.** Our approach is unfortunately still limited by some limitations of CLIP. Although we have good performance overall, as discussed in Sec. 4.2, our approach is hindered by the inability of CLIP to reason about relative relations in extremely fine-grained classification tasks. In addition, the entire approach is based on the premise that natural language descriptions can be helpful for image classification given enough context. Our experiments demonstrate that this is the case in general, but in exceedingly nuanced scenarios, the differences between different classes might be too intricate to be described by natural languages.

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
