# OpenReview forum: "ChatGPT-Powered Hierarchical Comparisons for Image Classification"
_NeurIPS.cc/2023/Conference — NeurIPS 2023 poster_

### Official Review · Reviewer_Y2ri · 2023-07-06

**Soundness:** 3 good
**Presentation:** 2 fair
**Contribution:** 2 fair
**Rating:** 5
**Confidence:** 4

**Summary:**

The article presents an approach for refining the classification performance of a vision-language model (i.e. CLIP), using descriptions derived from a large-language model (i.e. GPT-4). Different from previous attempts in this direction [26], directly extracting a discriminative description for each of the classes, this work proposes a hierarchical approach where more specific class descriptions are available at each layer. Starting from the class descriptions of [26], the text embeddings are used to perform clustering, creating groups of similar classes. Then, the LLM is used to obtain further descriptions to discriminate classes within each similar group. This process can be repeated iteratively until the number of classes in a group is low. Experiments on various benchmarks show that the proposed model consistently outperforms CLIP  and the direct descriptions of [26].

**Strengths:**

- The idea of splitting classification into multiple stages by asking for increasingly more specific descriptions is interesting as it allows the model to focus on easy separation first and increasingly more fine-grained ones later in the process. It also inherently gives interpretability, as we can check which type of descriptions lead to high scores for a certain class (Figure 4).

- Section 3.2 describe the type of query used, while the supplementary (Section 1.1) shows examples of failure cases of naive querying strategies.

- Despite the scores being very close among compared classes, Figure 4 provides a clear view of the type of descriptions involved in the classification process. Similarly, Figure 6 (despite being hard to interpret without the caption) shows the step-by-step changes in the model predictions, further complementing the info in Figure 5 on the hierarchical process.

- The framework is flexible, as it can be readily applied to new versions of large-language models and vision-language ones.

**Weaknesses:**

1. Despite consistently outperforming [26], the gap between the two models is relatively low: i.e. in Table 2, the average gain is always below 1% in accuracy. This comes unexpectedly as the model is initialized using descriptions extracted as in [26] (lines 134-136), and the article stresses the advantages of the proposed approach (e.g. captions of Fig. 1, lines 75-77,  lines 139-142, lines 171-174). Moreover, the proposed approach increases the computational cost, by increasing the number of text-image comparisons (i.e. the summation in Eq. 5). The claims on the performance/advantages should be revised to avoid overstating the contribution, and the computational cost discussed/reported.

2. The article reports limited analyses w.r.t. the method's hyperparameters. For instance, the model uses a factor $\lambda$ to balance the default CLIP prediction with the obtained hierarchical one (Eq. 6) but the impact of $\lambda$ is not analyzed. Similarly, this applies to the number of clusters $N$ and the size of the leaf nodes for the stopping criterion, $l$. It would also be interesting to show how the performance varies depending on the depth in the hierarchy, to confirm that deeper predictions lead to better performance.

5. $\tau$ is defined as the tolerance score to mitigate the above issue (Eq. 5, line 194). However, this value is set to 0 in the experiments (line 201). If $\tau$ is not used, it should not be defined, otherwise, it is important to specify the cases where it is used and analyze its impact.

3. [26] tests also on the EuroSAT dataset of satellite images, where providing natural language descriptions is challenging. This dataset is not included in the current experiments, but would further show the potential advantages of the approach in difficult classification settings.

4. Minor point: lines 185-194 describe the need to avoid that unrelated objects receive high scores for very specific descriptions. A strategy to avoid potential differences could be to just perform classification group-wise at each level of the hierarchy. This would work by "filtering" the classes by excluding every class that is not part of the predicted group for the next level of the hierarchy. Would this strategy be feasible? and in general, is there a reason for using the average (Eq. 8) rather than other aggregation mechanisms of the hierarchical scores?

**Questions:**

Following on the previous points:
1. Why is the gap between [26] and the proposed approach low?
2. What is the computational cost of the proposed model?
3. What is the impact of $\lambda$, and $N$?
4. How does performance vary w.r.t. the depth of the prediction?
6. Is $\tau$ used? If so, how does the performance vary?
5. What is the motivation behind using the average as the aggregator in Eq. 8?

Please, while answering, clarify also how the potential additional analyses/discussions would be included in the revised manuscript.

**Limitations:**

Section 5 discusses various limitations of the model, highlighting a fundamental one, i.e. the assumption that a category is easy to describe via natural language. Overall, I find the assessment of the limitations fair and thorough.

---

> ### Author Rebuttal · Authors · 2023-08-10
>
> We appreciate the reviewer's recognition of our approach's innovative concept of classification through specific descriptions, insights into our visualizations (Figs. 4, 5, and 6), and acknowledgment of the framework's adaptability to new large-language and vision-language models.
> ### [Q1] Improvement over [26]
> Thank you for highlighting the consistent improvement achieved across diverse datasets, which reinforces the efficacy of our approach. However, rather than pursuing comprehensive enhancements, we focus on refining semantic clarity within closely related classes. Hence, the extent of performance improvement varies based on the prevalence of challenging test samples in each dataset. Our results and visualizations compellingly validate the achievement of this objective through our hierarchical comparison approach.
> ### [Q2] Computational cost of the proposed model
> While our approach is indeed relatively more time-consuming than [26], it's important to note that the time-intensive aspect, constructing hierarchies, is a one-time operation for each dataset. Conversely, the subsequent stages, reasoning with CLIP models, are not slower than [26].
> - _Hierarchy construction_. Given the random latency of ChatGPT API calls, we report the time used to construct hierarchies in our experiments: 23 hours for ImageNet, 4 hours for CUB, 2.5 hours for Food101, 10 hours for Places365, 40 minutes for Oxford Pet, and 52 minutes for Describable Texture.
> Building hierarchies, while potentially time-consuming, is a one-time task per dataset.
> - _Reasoning_. The reasoning process has comparable runtime with our baseline [26]. The main difference is that we use Multi-level Score Fusion to aggregate scores, but since we implemented it in a parallelized way, the inference is about the same speed.
>
> In summary, even though our approach has a hierarchy construction procedure for each dataset, the overall efficiency is practical and acceptable.
> ### [Q3/4/5] Impact of hyperparameters and ablation study
> Thank you for your constructive suggestion! Ablation experiments are carried out to examine the impacts of $\lambda$, $\tau$, $N$, and prediction depth.
> - $\lambda$.  As illustrated in Tab. C, as $\lambda$ increases from 0 to 1, the mean progressively ascends from 61.25 to 62.01. This observed trend underscores a link between greater $\lambda$ values and higher accuracy. The peak accuracy of 62.12 is at $\lambda = 0.9$, prior to a marginal reduction to 62.01 at $\lambda = 1$.
> - $N$. $N$ determines the k-means algorithm's group count. Excessively large $N$ (e.g., 8) yields shallow hierarchies and coarse-grained descriptions; smaller $N$ prolongs hierarchy construction. An ablation on the ImageNet with ViT-B/32 for varying $N$ values follows. Notably, $N=4$ slightly improves accuracy at fourfold evaluation time. Conversely, $N=8$ accelerates sixfold but compromises by 1.64% in accuracy. Typically, $N$ falls within 4 to 6 and is proportional to the number of classes.
> | $N$ | Hierarchy Construction Time (Hours) | Accuracy (%) |
> |---|---|---|
> | 4 | 97 | 64.34 |
> | 6 | 23 | 63.88 |
> | 8 | 4 | 61.72 |
> - _The depth of the prediction_. The utilization of average aggregation facilitates score computation across different depths. The outcomes evaluated on the ImageNet with ViT-B/32, in the following table, highlight an enhancement in the second layer relative to the first, which is due to the introduction of contrasting descriptions. Subsequently, a consistent performance improvement is observed with increasing depth. The results are illustrated in Fig. 4 in the main paper.
> | Depth | Accuracy(%) |
> |---|---|
> | 1 | 57.91 |
> | 2 | 59.24 |
> | 3 | 59.57 |
> | 4 | 60.96 |
> | 5 | 61.72 |
> | 6 | 63.88 |
> - $\tau$. As shown in Tab. D, as $\tau$ increases from 0 to 1, the mean performance initially descends from 62.12 ($\tau=0$) to 61.95 ($\tau=0.6$), and then rebounding to 62.10 ($\tau=1$). The overall peak accuracy is 62.12 observed at $\tau = 0$, the value we use in the main paper.
> - $l$. As we stated in Line 201 (main paper), the threshold for leaf nodes is typically set to 2 or 3. Our observation indicates that ChatGPT tends to provide a less distinct differentiation of descriptions when handling more than 3 categories concurrently.
> - Across the majority of our ablation experiments, our approach consistently maintains commendable performance. These outcomes further underscore the robustness and efficacy of our approach, which remains steadfast even under diverse hyperparameter settings.
> ### [Q6] Motivation behind using the average aggregator in Eq. 8
> Thank you for asking. We employ the average as an aggregator for enhancing efficiency during inference. The inherent tree structure associates each class with distinct descriptions spanning varying levels of granularity. Thus, sequential representations manifest heterogeneity. The utilization of an average aggregator renders these features homogeneous, thereby significantly accelerating inference through parallelization.
>
> Our Multi-level Score Fusion maintains sequential arrangement across different levels, concurrently employing coarse-to-fine reasoning. The relevant code excerpt can be found in the Sup. Moreover, the effectiveness of our mean aggregator in capturing finer details with increasing layer depth is supported by Fig. 4 in the main paper.
> ### [Q7] Evaluation on EuroSAT
> Thank you for your suggestion. As shown in Tab. B, we evaluate our method on the EuroSAT dataset, which consistently attains an approximate 1% improvement.
> ### [Q8] Revision
> Thank you again for your constructive feedback! We plan to revise our manuscript (main paper) as follows:
> - Q1: Add a new paragraph in Sec. 4.2 after L242.
> - Q2: Add a new paragraph in Sec. 4.3 after L273.
> - Q3/4/5: Add new tables and paragraphs in Sec. 4.3 after L256.
> - Q6: Explain the motivation of average aggregation in detail after Sec. 3.3 at L185.
> - Q7: Add results to Tab. 2 and incorporate the analysis into the paragraph in Sec. 4.2 at L243.

---

> > ### Comment · Reviewer_Y2ri · 2023-08-14
> > **Thank you and further clarifications**
> >
> > I thank the authors for the thorough reply, answering the questions raised in all reviews. I especially appreciate the details on the computational cost and the provided analyses on the hyperparameters. For completeness, I would like to ask further questions/clarifications:
> > 1. According to Fig. 3, the depth of a class varies based on the clustering algorithm (please correct me if I am wrong). Given that various classes are found at different depth levels, how have the results above been computed? i.e. does depth N mean that all classifications for happen at depth N of the tree but if a leaf is found at node M<N, its value at M is used?
> >
> > 3. While the depth analysis shows a large improvement from 1 to 2 levels of the hierarchy, the description obtain by [26] achieve already a gain comparable to the depth 6 results (-0.4%) according to Table 2. This shows that asking for contrastive descriptions is less effective than asking for detailed ones if done in just a few rounds. Is this "issue" linked to the type of prompt used or to the clustering itself that gathers hard-to-contrast groups in the first levels?
> >
> > 2. The depth of the tree is proportional to the choice of the threshold value $l$. Will an explicit study on $l$ be included?
> >
> > 4. Are there failure cases in which an increase of depth in the decision leads to wrong predictions? If so it would be interesting to discuss them and why they happen.
> >
> > 2. Since $\tau$ is not used/does not have a positive impact on the final average results (according to Table D), will it still be included in the manuscript?

---

> > > ### Author Response · Authors · 2023-08-15
> > > **Thank You For the Feedback**
> > >
> > > ### Q1
> > > You are right that the depth of a class indeed fluctuates based on the clustering algorithm. In our experiment, when we utilize _all descriptions up to depth $N$_, if a leaf resides at depth $M$ in the final constructed tree, we consider two scenarios:
> > > - if $M < N$, we use descriptions from depth 1 to $M$;
> > > - if $M \ge N$, we use descriptions from depth 1 to $N$.
> > >
> > > ### Q2
> > > Very inspiring findings! We examine the hierarchy-building process and find that the groupings in the first few rounds are not suitable for direct classification. For example, in the CUB-200 dataset, at depth 1, 'Hooded Oriole', 'Scott Oriole', 'Baird Sparrow', 'Black-throated Sparrow', 'Chipping Sparrow', 'House Sparrow', and 'Grasshopper Sparrow' are grouped together, and summarized as "different sparrows". This phenomenon is due to the constant number of groups in the $k$-means algorithm, which can force diverse classes into a group and bias the summary towards the majority. As the grouping proceeds, diverse classes in the same group will eventually be separated, allowing for more reasonable comparisons. This also explains why our method will surpass [26] in later rounds.
> > >
> > > We admit that allowing $N$ to be adaptive would certainly make the hierarchy building more reasonable. However, since the clustering algorithm is not our main focus, we fix $N$ in the current method. Even though this will lead to crude miscategorization in the first few rounds, the grouping will be iteratively corrected and the overall classification performance consistently improved.
> > > ### Q3
> > > In the interest of time, we used the small "Texture" dataset (more complete experiments will be added in the final version) to do the experiment, and the results are as follows with the ViT-B/32 backbone. When $l$=3, it gets the best accuracy, and after that, performance will decrease as the $l$ increases.
> > > | $l$ | Accuracy |
> > > | -- | -- |
> > > | 2 | 47.86 |
> > > | 3 | **48.19** |
> > > | 4 | 47.23 |
> > > | 5 | 46.77 |
> > > | 6 | 46.19 |
> > > Our study's outcomes resonate with our initial intuition (L137-139, main paper) that it is difficult for ChatGPT to compare too many objects at once, and the answers obtained are often incomplete.  We deeply value your feedback and intend to integrate the findings and discussions into the supplementary material.
> > > ### Q4
> > > Based on the experimental results of the performance w.r.t the depth of the prediction, we can obtain confusion matrices based on different depths of the prediction. By comparing these confusion matrices, we find that there are indeed some classes whose classification accuracy decreases when using descriptions at greater depths. We pick representative classes and their descriptions of where they drop classification accuracy compared with only using the previous depth: [ND=newfoundland dog]
> > > - **Chow** (at 6th level):
> > >   - *"Coat color: NDs are typically black, brown, or gray, while chows can be red, black, blue, or cream."*
> > >   - *"Coat texture: NDs have a thick, shaggy double coat, while chows have a thick, woolly double coat that stands off from the body."*
> > >   - *"Facial features: chows have a distinctive "lion's mane" of fur around their necks and a wrinkled forehead, while NDs typically have a more uniform coat and less prominent facial wrinkles."*
> > >   - *"Size: NDs are generally larger than chows, with male NDs weighing up to 150 pounds and standing up to 28 inches tall, while chows typically weigh around 50-70 pounds and stand up to 20-25 inches tall."*
> > > - **Brown Bear** (at 5th level):
> > >   - *"has shaggy fur that varies from dark brown to light blonde."*
> > >   - *"is a large, broad head with small, round ears."*
> > >   - *"is a snout that is relatively short and broad compared to other bear species."*
> > >   - *"has curved, sharp claws that are ideal for digging and climbing trees."*
> > >
> > > We observe that when ChatGPT classifies similar classes at deeper levels, it uses attributes like length and size that are difficult to be interpreted in the image space. We experimented by replacing the vanilla CLIP prompts with these prompts and found that there is an increase of 10 to 20 error samples per class. Thus, deeper descriptions may be hard for CLIP to understand. Enhancing the alignment between LLMs and CLIP is indeed a promising avenue for future exploration. We greatly appreciate your suggestion and will incorporate related discussions into the supplementary material.
> > > ### Q5
> > > Your insight is highly valued, and we agree with your observation. Since $\tau$ has minimal influence on actual performance, we plan to omit it from equation (5) in the revised version of the paper. However, having observed that varying datasets might have slight improvements at different $\tau$ values, we plan to address this aspect in the supplementary material. This consideration ensures that our method remains both comprehensive and extensible while accommodating the nuanced effects of $\tau$ for different datasets. Your feedback has contributed greatly to refining our approach.

---

> > > > ### Comment · Reviewer_Y2ri · 2023-08-18
> > > >
> > > > I thank the authors for the additional clarifications. I have no further questions and I am increasing my score to borderline accept.
> > > >
> > > > However, I would like to suggest dedicating some space in the main paper for the analyses included in the rebuttal (after saving space by e.g. merging Tables 1 and 2). At the moment, the analyses on the model are qualitative (e.g. Fig. 4-6) and thus can provide only limited insight into the actual functioning of the approach. The rebuttal showed that a lot of interesting analyses can be provided, both quantitative (e.g. computational cost, prediction depth, hyperparameters) and qualitative (e.g. failure cases). Adding these insights would help future research on this topic.

---

> > > > > ### Author Response · Authors · 2023-08-18
> > > > > **Thanks for the reviewer**
> > > > >
> > > > > Thank you, your feedback is valued! We're thrilled to learn that our response effectively addressed your remaining questions. We will refine this paper based on your suggestions!

---

### Official Review · Reviewer_ufWY · 2023-07-07

**Soundness:** 3 good
**Presentation:** 3 good
**Contribution:** 3 good
**Rating:** 6
**Confidence:** 3

**Summary:**

This paper presents a hierarchical comparison method for zero-shot open-vocabulary image classification. By leveraging CLIP and Large Language Models (ChatGPT here), the authors build class hierarchies by recursively comparing and grouping classes. This hierarchy enables image classification by comparing image and text embeddings at different levels, improving accuracy and interpretability. The proposed method outperforms existing methods, achieving state-of-the-art performance on various datasets.

**Strengths:**

The proposed method exhibits high interpretability, as it draws inspiration from how human beings hierarchically organize visual and linguistic concepts in the real world. This intuitive approach enhances the understanding and explainability of the image classification process.
The efficient utilization of the extensive pretraining knowledge from ChatGPT/GPT-4 significantly improves the quality and granularity of the text descriptions.
The evaluations are conducted on 6 datasets and 4 different backbone settings, ensuring a robust assessment of the framework's effectiveness and generalizability.


**Weaknesses:**

It is worth considering recent open-source LLMs like LLaMA, Alpaca, and Vicuna for generating descriptions and conducting comparisons. These LLMs could provide alternative options and insights for the generation and evaluation of descriptions in the proposed framework.
The evaluation in the paper only includes one baseline method [26], which may limit the assessment of the proposed approach.
All the datasets used in the evaluation are in the general domain, which might not fully demonstrate the generalizability of the proposed method across various domains. Including datasets from diverse domains would further validate the effectiveness and robustness of the approach.

**Questions:**

Please see the weaknesses.

**Limitations:**

The utilization of ChatGPT/GPT-4 for generating descriptions incurs a cost, which could potentially restrict the reproducibility and impact of this work.

---

> ### Author Rebuttal · Authors · 2023-08-10
>
> We extend our heartfelt appreciation to you for acknowledging the high interpretability of our proposed method, which derives inspiration from human conceptual organization, and your recognition of the efficient utilization of ChatGPT's extensive pretraining knowledge and the comprehensive evaluations across multiple datasets and backbone settings.
>
> ### [Q1] Impact on the choice of large language models
>
> We deeply appreciate the valuable suggestion from the reviewer to explore alternative large language models beyond ChatGPT in our experiments. We illustrate the performance of our hierarchical comparison approach using a representative sample of Large Language Models (LLMs) in Fig. A.
>
> In all examined LLMs except for the small Multilingual Pre-Trained Transformer (MPT), the integration of descriptions resulted in performance gains that differed from the baseline. From a broader perspective, better LLMs usually lead to better descriptions and performance gains for CLIPs, and the relationship between LLMs' capability and CLIP accuracy is close to a linear correlation. However, the effect of the ChatGPT-based method jumps out of this linear relationship assumption and exhibits the most significant improvement. As a result, our approach shows efficacy in the vast majority of tested LLMs, and the description-generating power of ChatGPT is outstanding among them.
>
> ### [Q2] Datasets from special domains
>
> The outcomes derived from the EuroSAT dataset are showcased in Table B. Despite the distinctive nature of EuroSAT images being satellite-based and divergent from the majority of prevalent image classification datasets, our proposed approach manages to secure a substantial improvement of approximately 1% in performance across varied backbone architectures. This observation not only reaffirms the consistency of our findings as highlighted in the main paper's Tab. 2 but also underscores the adaptability and robustness of our approach across diverse imaging modalities.

---

> > ### Comment · Reviewer_ufWY · 2023-08-20
> >
> > Thank you for the feedback. All my concerns have been addressed, and I will keep the acceptance.

---

> > > ### Author Response · Authors · 2023-08-20
> > > **Thanks for the reviewer**
> > >
> > > Thank you for taking the time to review our work! We're pleased that our response successfully alleviated the concerns.

---

### Official Review · Reviewer_5GjG · 2023-07-07

**Soundness:** 3 good
**Presentation:** 4 excellent
**Contribution:** 3 good
**Rating:** 5
**Confidence:** 4

**Summary:**

This paper shows an example of introducing the power of LLMs in the zero-shot classification. This paper proposes a framework (or a process flow) rather than proposing a new mathematical algorithm. However, as shown in the experimental results section, it is demonstrated that the classification accuracy is about 1% point better than the state-of-the-art.

**Strengths:**

+ It is impressive to see how the integration of grouping and Language Model-generated descriptions to build knowledge trees can boost zero-shot classification performance. This is a compelling application of Large Language Models.

**Weaknesses:**

- I am afraid that the classification accuracy would depend on the performance of LLMs, but the detailed discussion on the accuracy of LLMs and its relation to the final classification performance is missing. It would also be interesting to compare multiple LLM models.
- I admit that the classification performance can be improved as can be seen in Tables 1 and 2. However, only the numbers are compared in those tables. The authors might want to discuss under what circumstances the proposed LLM-based one works better than the baselines and under what circumstances it does not. In other words, please discuss the remaining problems or limitations of the proposed method.

This is just a comment to improve the paper.
The texts in figures are too small, especially in Table 4. The authors might want to make them the same size as the text in the main body.


**Questions:**

Please see “Wekness” part.

**Limitations:**

Please see “Wekness” part.

---

> ### Author Rebuttal · Authors · 2023-08-10
>
> We extend our sincere gratitude to the reviewer for their recognition of our work's innovation in integrating grouping and Language Model-generated descriptions to enhance zero-shot classification performance.
>
> ### [Q1] Impact on the choice of large language models
>
> We deeply appreciate the valuable suggestion from the reviewer to explore alternative large language models beyond ChatGPT in our experiments. We illustrate the performance of our hierarchical comparison approach using a representative sample of Large Language Models (LLMs) in Fig. A.
>
> In all examined LLMs except for the small Multilingual Pre-Trained Transformer (MPT), the integration of descriptions resulted in performance gains that differed from the baseline. From a broader perspective, better LLMs usually lead to better descriptions and performance gains for CLIPs, and the relationship between LLMs' capability and CLIP accuracy is close to a linear correlation. However, the effect of the ChatGPT-based method jumps out of this linear relationship assumption and exhibits the most significant improvement. As a result, our approach shows efficacy in the vast majority of tested LLMs, and the description-generating power of ChatGPT is outstanding among them.
>
> ### [Q2] Remaining problems of the proposed method
>
> Please kindly refer to Section 2 of the supplementary material, specifically the subsection on "Failure Cases" where we delve into the types of descriptions that might adversely affect performance and pose challenges for the CLIP encoder.
>
> ### [Q3] Text size
>
> Thanks for pointing this out! We will enlarge the texts in the final version.

---

> > ### Comment · Reviewer_5GjG · 2023-08-14
> > **Thank you for the comments.**
> >
> > All of my concerns have been solved. I still keep on the acceptance side.

---

> > > ### Author Response · Authors · 2023-08-18
> > > **Thanks for the reviewer**
> > >
> > > We appreciate your diligent review! We're pleased that our response successfully tackled your remaining concerns.

---

### Official Review · Reviewer_bCYj · 2023-07-10

**Soundness:** 4 excellent
**Presentation:** 3 good
**Contribution:** 4 excellent
**Rating:** 7
**Confidence:** 4

**Summary:**

This paper argues that CLIP is limited due to the semantic ambiguity of similar classes. To overcome this, the paper proposes to construct a class hierarchy with large-language models and introduces a novel image classification framework based on hierarchical comparisons that compare embeddings at each level. Extensive experiments show promising results on multiple datasets under the zero-shot image classification task.

**Strengths:**

-This paper is well motivated. Using a class hierarchy to disambiguate the semantic embeddings of similar classes make a lot of sense.

-A novel hierarchical classification framework is proposed. To my knowledge, constructing a class hierarchy from LLMs is new. It is interesting to see this new application of LLMs. I also like the proposed plug-and-play hierarchical classification method which does not need to retrain the CLIP model.

-Empirically, it consistently improves the CLIP and Open-CLIP on 6 datasets under the zero-shot image classification task. The interpretability is another interesting property of the method.

**Weaknesses:**

-Comparisons between the class hierarchy generated by LLMs and the ones labeled by humans e.g., WordNet are missing.

-What is the impact of the hyperparameters, or example, the threshold to construct the hierarchy?

**Questions:**

I do not have additional questions at this stage.

**Limitations:**

Limitations are discussed in the paper.

---

> ### Author Rebuttal · Authors · 2023-08-09
>
> We express our gratitude to the reviewer for their valuable insights and recognition of our motivation, innovative hierarchical classification framework, and the significance of our experimental results and interpretability. In the subsequent sections, we address the concerns raised by the reviewers.
>
> ### [Q1] Comparison with WordNet
>
> We are grateful for your attention to WordNet, as it offers pre-defined descriptions for a wide variety of classes.
>
> However, one primary emphasis of our research centers around the generation of visually comparative descriptions. This emphasis stems from our utilization of the CLIP model, which aligns the feature space of image and language. On the contrary, WordNet descriptions
>
>   - subsist in isolation, devoid of direct interdependencies with other definitions (similar to [26]), and
>   - often lacks any provision for visual descriptions.
>
> Therefore, we argue that, compared to our hierarchical and comparative descriptions, WordNet descriptions are less suitable for zero-shot open-vocabulary image classification. This elucidates our rationale for excluding WordNet-based descriptions from the initial submission.
>
> Empirically, we gather WordNet descriptions for each category and append them to the original text, and evaluate the performance across two distinct model configurations (ResNet-50 and ViT-B/32) by computing the average accuracy. The corresponding outcomes are tabulated in Tab. A. Consistent with our analysis, the employment of WordNet-based descriptions exhibits a performance degradation compared to the inherent effectiveness of the CLIP default prompt, "a photo of {class_name}".
>
> ### [Q2] Impact of hyperparameters
> Thank you for your constructive suggestion! Ablation experiments are carried out to examine the impacts of $\lambda$, $\tau$, $N$, and prediction depth.
> - $\lambda$.  As illustrated in Tab. C, as $\lambda$ increases from 0 to 1, the mean progressively ascends from 61.25 to 62.01. This observed trend underscores a link between greater $\lambda$ values and higher accuracy. The peak accuracy of 62.12 is at $\lambda = 0.9$, prior to a marginal reduction to 62.01 at $\lambda = 1$.
> - $N$. $N$ determines the k-means algorithm's group count. Excessively large $N$ (e.g., 8) yields shallow hierarchies and coarse-grained descriptions; smaller $N$ prolongs hierarchy construction. An ablation on the ImageNet with ViT-B/32 for varying $N$ values follows. Notably, $N=4$ slightly improves accuracy at fourfold evaluation time. Conversely, $N=8$ accelerates sixfold but compromises by 1.64% in accuracy. Typically, $N$ falls within 4 to 6 and is proportional to the number of classes.
> | $N$ | Hierarchy Construction Time (Hours) | Accuracy (%) |
> |---|---|---|
> | 4 | 97 | 64.34 |
> | 6 | 23 | 63.88 |
> | 8 | 4 | 61.72 |
> - _The depth of the prediction_. The utilization of average aggregation facilitates score computation across different depths. The outcomes evaluated on the ImageNet with ViT-B/32, in the following table, highlight an enhancement in the second layer relative to the first, which is due to the introduction of contrasting descriptions. Subsequently, a consistent performance improvement is observed with increasing depth. The results are illustrated in Fig. 4 in the main paper.
> | Depth | Accuracy(%) |
> |---|---|
> | 1 | 57.91 |
> | 2 | 59.24 |
> | 3 | 59.57 |
> | 4 | 60.96 |
> | 5 | 61.72 |
> | 6 | 63.88 |
> - $\tau$. As shown in Tab. D, as $\tau$ increases from 0 to 1, the mean performance initially descends from 62.12 ($\tau=0$) to 61.95 ($\tau=0.6$), and then rebounding to 62.10 ($\tau=1$). The overall peak accuracy is 62.12 observed at $\tau = 0$, the value we use in the main paper.
> - $l$. As we stated in Line 201 (main paper), the threshold for leaf nodes is typically set to 2 or 3. Our observation indicates that ChatGPT tends to provide a less distinct differentiation of descriptions when handling more than 3 categories concurrently.
> - Across the majority of our ablation experiments, our approach consistently maintains commendable performance. These outcomes further underscore the robustness and efficacy of our approach, which remains steadfast even under diverse hyperparameter settings.

---

> > ### Comment · Reviewer_bCYj · 2023-08-18
> >
> > I would like to thank the authors for the response. My concerns are all properly addressed. I think the rebuttal addressed most of the concerns of other reviewers too. Therefore, I would keep my rating as an Accept.

---

> > > ### Author Response · Authors · 2023-08-18
> > > **Thanks for the reviewer**
> > >
> > > Thanks for your reviewing efforts! We're glad that our rebuttal was able to address your remaining concerns!

---

### Author Rebuttal · Authors · 2023-08-10

We would like to express our sincere gratitude to all the reviewers for their invaluable time and expertise invested in the meticulous review of our paper. Your thoughtful feedback has immensely contributed to refining our work. We are also encouraged by the recognition of our paper's strengths:

- Reviewer `bCYj`: The well-motivated approach, novel hierarchical classification framework, empirical effectiveness, and interpretability.
- Reviewer `5GjG`: Integration of grouping and language-model-generated descriptions and compelling application of large language models.
- Reviewer `ufWY`: High interpretability, efficient utilization of pretraining knowledge, and comprehensive evaluations.
- Reviewer `Y2ri`: Staged classification approach, inherent interpretability, comprehensive content, and flexibility of the framework.

These insights resonate with our aim to present a training-free, intuitive, effective, and explainable approach for zero-shot open-vocabulary image classification. We are deeply committed to incorporating all the valuable feedback received and refining our manuscript accordingly in the revised version.

---

### Decision · Program_Chairs · 2023-09-21

**Decision:**

Accept (poster)

**Comment:**

The paper proposes a method to improve classification with CLIP using LLMs to create a discriminative class hierarchy, and an algorithm top use this hierarchy to create classification decisions. The reviews agree that the method is well motivated, interpretable, and novel, plus the results are convincing. The scores are unanimous in recommending acceptance. Concerns were raised about the depth of the analysis and ablation, but these have been addressed in the rebuttal. Therefore, I recommend acceptance. The authors should include the data presented in the responses to reviewers `bCYj` and `Y2ri` in the camera ready version of the paper.